# Age of Cafeteria Diet Onset Influences Obesity Phenotype in Mice in a Sex-Specific Manner

**DOI:** 10.3390/ijms252212436

**Published:** 2024-11-19

**Authors:** Nadezhda Bazhan, Antonyna Kazantseva, Anastasia Dubinina, Natalia Balybina, Tatiana Jakovleva, Elena Makarova

**Affiliations:** The Laboratory of Physiological Genetics, The Institute of Cytology and Genetics, 630090 Novosibirsk, Russia; antonyna-sh@yandex.ru (A.K.); dubinina_anasta@mail.ru (A.D.); n.balybina@alumni.nsu.ru (N.B.); jakov@bionet.nsc.ru (T.J.); enmakarova@gmail.com (E.M.)

**Keywords:** age, sex, diet-induced obesity, FGF21, mice, gene expression

## Abstract

We investigated the influence of sex and the age of obesogenic diet initiation on the obesity phenotypes at a later age. C57Bl mice started the Cafeteria Diet (CafD, with increased fat and carbohydrates, ad libitum, from 7 weeks of age (7CafD, pre-puberty) or 17 weeks of age (7CafD, post-puberty) while control C57Bl mice were fed regular chow. At 27 weeks of age, 7CafD males (n = 9) compared to 17CafD males (n = 7) had lower body weight, white adipose tissue (WAT) relative weight, and plasma cholesterol levels, and a higher expression of thermogenic genes in WAT and brown adipose tissue (BAT), and fatty acid oxidation (FAO) and insulin signalling genes in muscles. The 7CafD females (n = 8), compared to 17CafD females (n = 6), had higher plasma triglyceride levels and hepatic glycogen content, but lower insulin sensitivity and hepatic expression of FAO and insulin signalling genes. The 7CafD females, compared to 7CafD males, had more WAT, and a reduced expression of FAO genes in muscles and thermogenic genes in WAT. The 17CafD females, compared to 17CafD males, had lower plasma leptin and insulin levels, and higher insulin sensitivity and expression of insulin signalling genes in the liver and muscles. Thus, the initiation of the obesogenic diet before puberty led to a more adaptive metabolic phenotypes in males, and after puberty, in females.

## 1. Introduction

The prevalence of obesity is increasing not only in adults, but also in adolescents and even children. Prepubertal and early post-pubertal ages in humans and animals differ significantly from adult age in a number of metabolic parameters. We have previously shown that when male mice are fed a standard diet, the metabolic profile differs between young (10 weeks of age) and early adult (15 weeks of age) mice. At the whole organism level, age differences were primarily reflected in blood and testicular testosterone levels, which were many times higher in 15-week-old male mice than in 10-week-old males [1]. High blood testosterone concentrations in 15-week-old males were associated with a higher expression of muscle lipid oxidation genes and genes involved in lipolysis and glucose transport in white adipose tissue (WAT) [2]. However, it is unknown whether these age-related differences in metabolic phenotype affect the rate of obesity and the mechanisms of adaptation to high calorie food intake. 

The metabolic phenotype Is not only dependent on the age of the animals. Sex is known to influence energy metabolism in both standard chow [3] and obesogenic diets [4,5]. We have reported that male and female C57Bl/6J mice differ in their adaptation to a cafeteria diet (CafD), a diet containing an increased amount of fats and carbohydrates [5]. In females, CafD induced greater adiposity than in males and downregulated the expression of genes related to lipogenesis and glucose metabolism in the liver. In contrast, in males, CafD increased lean mass gain, induced hyperinsulinemia, and upregulated the expression of genes involved in lipid and glucose oxidation in WAT and liver, and lipogenesis and lipolysis in WAT, suggesting reduced energy expenditure.

Fibroblast growth factor 21 (FGF21), a liver hormone, is known to regulate adaptation to metabolic stress, including high-calorie diets that induce obesity [6]. In animals on standard chow, the activity of the FGF21 system is very low, but the consumption of sweet and fatty foods activates the FGF21 system and increases blood levels of FGF21 hundreds of times [5]. We have previously shown that there are sex differences in the activity of the FGF21 system in obese animals with the long-term consumption of sweet and fatty foods: in males, FGF21 expression in the liver, brown adipose tissue, and blood FGF21 levels were higher than in females [5].

The susceptibility to obesity changes during life in a sex-specific way [7,8]. Salinero et al. [9] showed that adaptation to a high-fat, obesogenic diet in mice was age and sex dependent. When the HFD was initiated in young mice (6 weeks), females gained less weight and maintained better glucose tolerance than males. When the diet was started in early adulthood (9 weeks), no sex differences in adaptation were observed. When the diet was initiated in middle-aged mice (32 weeks), males had an advantage in adaptation. These data indicate that sex differences in the DIO effect vary with age at diet initiation, highlighting the importance of both age and sex as biological variables in research [9]. However, the mechanisms of adaptation to high-calorie foods in males and females at different ages remain unexplored. Therefore, the aim of this work is to compare the metabolic phenotype and adaptation strategies of adult male and female mice fed an obesogenic diet at different ages.

## 2. Results

### 2.1. Lipid Metabolism

Obesity developed in males fed the cafeteria diet (CafD) from different ages (Figure 1). Body weight, white adipose tissue (WAT), and hepatic triglyceride (TG) content, as well as metabolic and hormonal blood parameters, were higher than in controls (except for blood free fatty acid levels). The age at which the diet was started influenced lipid metabolism: body weight, relative white adipose tissue (WAT) weight, leptin, adiponectin, and blood cholesterol levels were lower in 7CafD males than in 17CafD males. In females maintained on both CafD diets, body weight and relative WAT weight, hepatic TG content, blood concentrations of cholesterol, and leptin were higher, and relative liver weight was lower, than in the control. Blood levels of fatty acids, adiponectin, and TGs were not different from the control. Indicators of lipid metabolism in females did not depend on the age at which the diet was started. The exception was the blood level of TGs, which was higher in the 7CafD females than in the 17CafD females.

In females, BW in all groups and plasma leptin levels in the 17CafD group were lower than in males, and relative WAT weight in the 7CafD group was higher than in males.

The transcriptional response of adipose tissue suggests that the accumulation of white fat in obese males was associated with a decrease in the intensity of lipid metabolism. In WAT, the expression of most genes related to fatty acid oxidation (*Ppargc1* and *Pparγ*), lipolysis (*Lipe* and *Pnpla2*), and lipogenesis (*Fasn*) was lower in obese males than in controls (Figure 2). The decrease in *Fasn* expression relative to control did not depend on the age at which the diet was started, while the decrease in the expression of fatty acid oxidation and lipolysis genes was less pronounced in 7CafD males than in 17CafD males. As a result, in WAT, the expression of *Ppargc1*, *Pparγ*, *Lipe*, and *Pnpla2* was higher in 7CafD males than in 17CafD males.

In obese females, the expression of *Ppargc1*, *Fasn*, *Lipe,* and *Pnpla2* was lower in WAT than in the control. The decrease in *Fasn* and *Ppargc1* expression was independent of the age at which the diet was started, and the decrease in lipolysis gene expression was less pronounced in 7CafD females than in 17CafD females. Thus, the mRNA level of lipolysis genes was higher in 7CafD females than in 17CafD females. In obese females, *Cpt1α* expression was not significantly different from controls, but it was higher in 7CafD females than in 17CafD females. 

Only in the control group was the expression of all WAT genes involved in lipid oxidation and lipolysis lower in females than in males.

In obese males, increased hepatic TG accumulation compared to controls was associated with an increased expression of genes regulating fatty acid oxidation (*Pparα*, *Cpt1α,* and *Ppargc1*) (Figure 3). The increase in *Pparα* expression did not depend on the age at which the diet was started, while the expression of *Cpt1α* and *Ppargc1* was increased relative to the control only in 17CafD males (*Cpt1α*—significant and *Ppargc1*—tendency, *p* < 0.06). The expression of genes involved in the synthesis of fatty acids and lipoproteins did not differ from the control (*Fasn*, *Acca*) or was lower than the control (*Accb*) and did not depend on the age at which the diet was started. In 7CafD males, the decrease in *Accb* expression was at the level of a trend (*p* < 0.07) and in 17CafDC males it was significant. In females, TG accumulation in the liver was associated with an increased expression of genes involved in fatty acid oxidation and lipoprotein synthesis compared to the control only in 17CafD females.

Only in the 17CafD group was the expression of hepatic genes involved in the oxidation (*Ppargc1*) and synthesis (*Accβ*) of fatty acids higher in females than in males. 

In male skeletal muscle, the transcriptional response to the diet depended on the age at which the diet was started: in 7CafD males, the expression of fatty acid oxidation genes (*Cpt1b* and *Ucp3*) was higher than in control and 17CafD males (Figure 3). In females, the expression of muscle genes involved in fatty acid oxidation did not differ from the control and did not depend on the age at which the diet was started.

Only in the 7CafD group was the expression of muscle genes involved in fatty acid oxidation (*Cpt1β* and *Ucp3*) lower in females than in males.

### 2.2. Carbohydrate Metabolism

CafD males developed hyperinsulinemia. This was independent of the age at which the diet was started (Figure 4). Blood glucose levels in 7CafD males were not different from those in controls, but were higher in 17CafD males than in controls. Liver glycogen content in obese males was not different from that in control males. The transcriptional response in liver and WAT indicates the development of insulin resistance in obese mice. Obese males had a decreased expression of the insulin receptor substrate type 2 gene *(Irs2*) in the liver and the insulin receptor gene (*Insr*) and glucose transporter gene (*Slc2a4*) in the WAT. In addition, the expression of the *Pck1* gene (an indicator of gluconeogenesis) was decreased and the expression of the *Pklr* gene (an indicator of glycolysis) was increased in the liver of obese males (Figure 5). In muscle, the expression of *Irs1*, *Irs2,* and *Slc2a4* was downregulated in 17CafD males, whereas the expression of *Insr* and *Slc2a4* was upregulated in 7CafD males. Age at the start of the diet influenced gene expression in WAT (*Insr*, *Slc2a4*) and muscle (*Insr*, *Irs1*, *Slc2a4*): it was higher in 7CafD males than in 17CafD males.

Obese females had elevated blood insulin and glucose levels, indicating the development of insulin resistance, which was more pronounced in 7CafD females (Figure 4). In 7CafD females, blood insulin levels and liver glycogen content were higher than in 17CafD females. Insulin resistance in 7CafD females was associated with a reduced expression of insulin signalling and glucose transport genes in all metabolic tissues: *Insr* and *Irs2* in liver, *Insr* and *Slc2a4* in WAT, and *Slc2a4* in muscle. In 17CafD females, the expression of *Insr* and *Slc2a4* in WAT and *Slc2a4* in muscle was also reduced, while in the liver the expression of Insr and *Irs2* was either not different from the control or even higher than in the control females (*Irs1*).

As a result, the expression of hepatic *Insr* and *Irs1* was lower in 7CafD females than in 17CafD females. In obese females of both groups, the expression of *Pklr* (an indicator of glycolysis) was higher (for 7CafD females, trend *p* < 0.07) and *Pck1* (an indicator of gluconeogenesis) was lower than in control animals (Figure 5).

In obese mice, only in the 17CafD group, carbohydrate metabolism indices were more favourable in females than in males: they had a lower blood insulin level compared to males (Figure 4), increased insulin sensitivity at the whole organism level and at the level of expression of genes related to insulin signalling in the liver (*Irs2*) and in the muscles (*Irs1* and *Irs2*) (Figure 5).

### 2.3. Thermogenesis

Only in 17CafD males was BAT mass significantly higher than in the control (Figure 6). The transcriptional response to the diet indicated an activation of thermogenesis only in 7CafD males, in which the expression of *Ucp1* was higher than in the control in BAT and the expression of *Ucp1* and *Dio2a* in WAT. In addition, 7CafD males had a higher expression of genes involved in fatty acid oxidation (*Pparγ* and *Cpt1b*) in BAT than 17CafD males. In obese females, BAT mass was increased compared to controls, regardless of the age at which the diet was started.

The transcriptional response to the diet indicated a suppression of thermogenesis, which was less pronounced in 7CafD females. In 7CafD females, only *Dio2* expression was decreased in WAT and BAT, whereas in 17CafD females *Ucp1*, *Dio2,* and *Pparγ* expression was decreased in BAT and *Dio2* expression was decreased in WAT.

Only in the control group, *Ucp1* expression in BAT was higher in females than in males.

### 2.4. FGF21 System Activity

The activity of the FGF21 system was assessed by measuring blood FGF21 levels and the tissue expression of *Fgf21* and *Klb* (an indicator of sensitivity to FGF21 [6]). In males, dietary intake increased blood FGF21 levels to a greater extent in 7CafD males: the difference from the control was significant (Figure 7). The activation of *Fgf21* expression in liver and BAT depended on the age of diet initiation: *Fgf21* mRNA levels were higher in 7CafD males than in 17CafD males. *Klb* expression in liver was similar, in muscle it was higher, and in WAT and BAT it was lower in obese males compared to controls, the effect of the age of diet initiation on FGF21 sensitivity was found only in BAT: *Klb* mRNA levels were higher in 7CafD males than in 17CafD males.

In females, dietary intake increased blood levels of FGF21. The hormone level in the blood of 7CafD females was three times higher than that of 17CafD females (trend *p* < 0.06). Increased blood levels of FGF21 were associated with an increased expression of the *Fgf21* gene in the liver, WAT, and BAT. In obese females, *Fgf21* expression in the liver was lower than in obese males, regardless of the age at which the diet was started. Age at diet initiation affected *Fgf21* expression only in BAT: 7CafD females had higher *Fgf21* mRNA levels than 17CafD females. In females, the effect of diet on *Klb* expression was found only in 17CafD females: their *Klb* mRNA level was higher in the liver and lower in the BAT than in the control. In addition, hepatic *Klb* expression was lower in 7CafD females than in 17CafD females. 

## 3. Discussion

In this work, we investigated the mechanisms of adaptation to an obesogenic diet started at different ages in male and female mice. We compared the absolute metabolic rates of males and females and their adaptation strategy.

In males, an early initiation of the diet contributed to a more effective adaptation. At the end of the experiment, at 27 weeks of age, early dieting males (7CafD) had lower body and white adipose tissue (WAT) weights and lower blood leptin, adiponectin, and cholesterol concentrations than late dieting males (17CafD). The lower WAT deposition in 7CafD males compared to 17CafD males could be related to more intense fatty acid oxidation (FAO) and lipolysis in WAT. This hypothesis is supported by the data on the higher expression of lipolysis (*Lipe* and *Pnpla2*) and FAO (*Ppargc1* and *Pparγ*) genes in 7CafD males compared to 17CafD males. In addition, the differences in body weight and WAT weight could be the result of differences in energy expenditure, which is mainly determined by thermogenesis in adipose tissue and FAO in brown adipose tissue (BAT) and muscle. Thermogenic gene expression indicated that thermogenesis was higher in 7CafD males than in 17CafD males: only 7CafD males had an increased expression of *Ucp1* in WAT and BAT and *Dio2* in WAT compared to controls. Furthermore, only 7CafD males had significantly higher muscle expression of FAO genes (*Ucp3* and *Cpt1b*) compared to controls. We have previously shown that the level of *Ucp3* and *Cpt1b* expression in the muscles of male mice fed standard chow depends on the age of the animal: in juveniles, the expression of these genes was lower than in early adulthood [2]. It can be assumed that the initially high ‘basal’ level of expression did not allow it to increase further during the process of adaptation to CafD.

Obesity is known to be associated with reduced insulin sensitivity. Blood insulin levels in obese males were many times higher than in control males, irrespective of the age at which the diet was started and the degree of obesity, but hyperglycaemia only developed in 17CafD males. The age at which the diet was started influenced the expression of genes involved in insulin signalling. In 7CafD males, the expression of *Irs2* was reduced in liver and the expression of *Insr* and *Slc2a4* was reduced in WAT, but in muscle it was either not different from control (*Irs1* and *Irs2*) or higher (*Insr* and *Slc2a4*). Overall, the expression of insulin signalling genes in muscle and glucose transporter genes in muscle and WAT was higher in 7CafD males than in 17CafD males.

The hepatokine FGF21, whose expression increases with sweet food consumption and obesity, and which can have endocrine and autocrine effects, acts as an adaptation factor [10,11]. FGF21 acts to restore homeostasis by coordinating metabolic responses from WAT, BAT, muscle, liver, and hypothalamus [12,13,14]. The effect of FGF21 on the metabolic phenotype is mediated in part by its effect on gene expression involved in FAO, glucose metabolism in the liver and glucose transport and FGF21 activity in WAT [15,16], and FAO genes in muscle [17]. We assessed the activity of the FGF21 system by plasma FGF21 levels and the tissue expression of the *Fgf21* gene itself and the *Klb* coreceptor (an indicator of sensitivity to FGF21 [6]). In males, dietary intake increased blood levels of FGF21 and its gene expression in all tissues. Blood FGF21 levels, and *Fgf21* mRNA levels in liver and BAT, were significantly higher in 7CafD males than in 17CafD males. Whether there is systemic FGF21 resistance in obese animals is controversially discussed in the literature [18,19].

It should be noted that in 7CafD males, the reduction in *Klb* expression in the BAT was less pronounced than in 17CafD males, and expression was higher than in the latter. Given the higher expression of *Fgf21* itself, it can be speculated that the activity of the local FGF21 system was higher in 7CafD males than in 17CafD males, and that endocrine or autocrine effects of FGF21 could determine the activity of thermogenesis in BAT. In contrast to adipose tissue, *Klb* expression in muscle, and therefore sensitivity to FGF21, was increased in obese males. The pharmacological administration of FGF21 is known to increase muscle FAO gene expression [17] and insulin sensitivity [20,21,22]. As circulating FGF21 levels were almost twice as high in 7CafD males as in 17CafD males, it is possible that the upregulation of genes involved in FAO, insulin signalling, and muscle glucose transport is due to the endocrine effects of FGF21, but many other mechanisms not investigated in our work may also be involved.

Females adapted differently to the consumption of sweet and fatty foods than males: the age at which the diet was started had no effect on the severity of obesity and hyperglycaemia at the end of the experiment. However, blood insulin levels in 7CafD females were almost three times higher than in 17CafD females, suggesting that insulin sensitivity in 7CafD females was significantly lower than in 17CafD females. This suggestion is supported by the fact that 7CafD females had elevated liver glycogen and blood TG levels compared to 17CafD females. High blood TG levels are a marker of insulin resistance, as insulin normally promotes the uptake of TG from the blood [23]. Although 7CafD females had elevated blood TG levels, hepatic TG levels did not differ from those of 17CafD females. Hepatic TG levels are determined not only by insulin-dependent uptake from the blood, but also by the fat content of the diet. The deposition of white fat in the depot did not differ between 7CafD and 17CafD females. We suggest that excess dietary fat intake promotes liver deposition equally in 7CafD and 17CafD females.

Taken together, these data suggest that 7CafD females have lower insulin sensitivity than 17CafD females. This is supported by data on the expression of insulin signalling and glucose transport genes. In 7CafD females, the expression of these genes was reduced in all tissues compared to the control: *Insr* and *Slc2a4* in WAT, *Slc2a4* in muscle, and *Insr* and *Irs2* in liver. In 17CafD females, the expression was reduced in WAT and muscle, but not in liver (*Insr* and *Irs2*) or was higher than in control females (*Irs1*). Thus, in the liver, the expression of all genes related to insulin signalling was higher in 17CafD females than in 7CafD females.

Gene transcription profiles indicate that in obese females, regardless of the age of diet initiation, changes in thermogenesis and lipid metabolism in adipose tissue and muscle were non-adaptive and reflected the deleterious effects of the diet. Adaptive changes were found only in the liver of 17CafD females, where the diet significantly increased FAO gene expression (*Pparα*, *Cpt1α*, and *Ppargc1*), the mRNA levels of these genes being significantly higher than in 7CafD females.

In males, the FGF21 system appeared to be one of the factors determining adaptation to diet. In females, the effect of diet on the activity of the FGF21 system was generally similar to that in males: FGF21 levels in the blood, *Fgf21* expression in the liver, and BAT were more pronounced in 7CafD females than in 17CafD females. Was the FGF21 system involved in dietary adaptation in females? According to our data, in contrast to males, higher blood FGF21 levels and increased *Fgf21* expression in liver and BAT in 7CafD females were not associated with decreased WAT deposition or an increased expression of genes involved in the regulation of thermogenesis in BAT. Furthermore, we suggest that 7CafD females had a less efficient adaptation than 17CafD females. In 7 CafD females, insulin sensitivity, expression of hepatic genes related to insulin signalling and FAO were lower than in 17 CafD females, despite the fact that the activity of the FGF21 system was higher in them than in 17 CafD females. It has been shown that activation of fatty acid oxidation at peroxisomes in the liver enhances thermogenic responses [24], i.e., increases energy expenditure. The activation of hepatic FAO gene expression suggests that in 17CafD females the mechanism of adaptation to the diet could be realised by an increase in energy expenditure in the liver.

The activation of hepatic FAO gene expression in 17CafD females could not be a consequence of an autocrine, endocrine, or central influence of FGF21: firstly, the liver is not a target organ of FGF21, since the level of expression of FGFR1 receptors in the liver is very low [25]; secondly, the level of circulating FGF21 in 17CafD females is lower, albeit not significantly, than in 7CafD females. We found a significant upregulation of the expression of the co-receptor *Klb* in the liver of 17CafD females. It is known that the protein it encodes, in complex with FGFR4, mediates the influence of another endocrine growth factor, FGF15, which regulates bile acid metabolism and insulin sensitivity [26,27]. It is possible that the activation of hepatic genes involved in FAO and insulin signalling in 17CafD females is related to the influence of FGF15. It has been shown that in leptin-deficient *ob/ob* mice given a lipid emulsion injection, FGF19 (an analogue of FGF15 in humans) increased FFA excursion, reduced respiratory quotient and lowered serum triglyceride [28]. 

It is interesting to note that an early or late age of diet initiation not only determined different adaptation strategies in each sex, but also altered the manifestation of sex dimorphism in adaptation. The 7CafD females developed a less favourable obesity phenotype than 7CafD males: they accumulated more WAT, and exerted a reduced expression of FAO genes (*Cpt1β* and *Ucp3*) in muscle and of the thermogenic gene (*Dio2*) in WAT. These data suggest that greater adiposity in 7CafD females was due to reduced lipid oxidation in muscle and thermogenesis in WAT. In the 17CafD group, the opposite situation is observed: the obesity phenotype was more favourable in females than in males: they had lower blood leptin and insulin levels, signs of higher insulin sensitivity, both at the whole body level and at the level of expression of genes related to insulin signalling in the liver (*Irs2*) and muscles (*Irs1* and *Irs2*).

Thus, we have shown for the first time that the age of onset of the diet not only influences the adaptation strategy in each sex, but also the manifestation of the sex dimorphism in the obesity phenotype: early onset of the diet was associated with a more favourable phenotype in males and later onset of the diet was associated with a more favourable phenotype in females. The physiological mechanisms underlying this phenomenon are unknown. They are thought to be related to the dynamics of sex hormone activity during normal development. Both testosterone and oestradiol are known to regulate many aspects of energy metabolism [29,30,31]. In particular, the beneficial effects of oestradiol on insulin sensitivity in the liver are well established [32,33,34]. The number of oestrogen receptors in the liver has been shown to increase with reproductive maturity [35]. This suggests that the effects of oestrogen on liver function may be greater in 17CafD females, where dietary adjustment began after puberty, than in 7CafD females, where dietary adjustment began before puberty, and in 17CafD males, who are deprived of the beneficial effects of oestrogen.

The main limitation of our work is the lack of data on plasma levels of sex hormones and growth hormone. This information would help to understand the mechanisms of differential adaptation to diet in prepubertal and adult mice of different sexes. The small number of animals in certain subgroups for statistical analyses may also be a limitation.

CafD is one of the models for the development of diet-induced obesity (DIO) in humans. Many studies on the mechanisms of DIO have been carried out in mice [20,36,37] and rats [38] that started consuming the diet before puberty, which essentially models the development of childhood obesity. The results of our work indicate that the conclusions drawn from studying the influence of diet in juvenile animals cannot be applied to explain the mechanisms of adaptation in adult animals, and that the results obtained in males cannot be extended to females. Further research is needed to identify the mechanisms by which sex and age of diet onset interact in adaptation to DIO.

## 4. Materials and Methods

### 4.1. Animals and Experimental Design 

The study was conducted in accordance with the tenets of the Declaration of Helsinki and approved by the Independent Ethics Committee of the Institute of Cytology and Genetics, Siberian Branch, Russian Academy of Sciences (protocol number No 76, 7 April 2021).

Experiments were performed with C57BL/6J mice housed in the vivarium of the Institute of Cytology and Genetics, Novosibirsk, Russia. Animals were maintained on a 12 h light/12 h dark cycle with free access to food and water. Male and female mice were placed 3 to a cage. Mice of each sex were divided into three groups: mice of the first group (7 males and 6 females) received standard food, standard pelleted chow (control group), and the second and third groups received obesogenic food, standard food in combination with lard and sweet butter cookies. A diet with an increased amount of fat and carbohydrates, the so-called “cafeteria diet” (CafD), mimics the diet that causes the development of obesity in humans [39]. Mice in the second group (pre-pubertal, 9 males and 8 females) started the diet at 7 weeks of age (7CafD group) and finished the diet after 20 weeks. Mice in the third group (post-puberty, 6 females and 7 males) started the diet at 17 weeks of age (17CafD group) and finished the diet after 10 weeks. This means that the mice in the second and third groups differed in the age at which they started the diet (young and early adult) and the duration of the diet (20 and 10 weeks); they finished the diet at the same age of 27 weeks. Mice at this age were sacrificed by decapitation, and blood and tissue samples were collected for biochemical analysis and gene expression assessment. Liver, total visceral white adipose tissue (WAT), and interscapular brown adipose tissue (BAT) were weighed. Gene expression was measured in samples of liver, WAT (paragonadal location), BAT, and skeletal muscle *Musculus quadriceps femoris.*

### 4.2. Diets

The standard diet was purchased from BioPro, Novosibirsk, Russia. Ingredients: Two-component cereal mix, dairy ingredients, protein-rich ingredients (vegetable and animal proteins), vegetable oil, amino acids, organic acids, vitamin-mineral premix, and fibre. Crude protein: 22%. Energy value: 250 kcal/100 g. CafD contained, in addition to the standard granules, pork lard and biscuits purchased from a grocery store. The composition of the biscuits (g/100 g): proteins—6.9, fats—18.4, and carbohydrates—71.8. Energy value: 458 kcal/100 g. Lard (subcutaneous fat): proteins—1.8, fats—94.2, and carbohydrates—0. Energy value: 800 kcal/100 g.

### 4.3. Plasma Assays and Triglyceride and Glycogen Measurements

Concentrations of insulin, leptin, and adiponectin were measured using Rat/Mouse Insulin ELISA Kit, Mouse Leptin ELISA Kit (EMD Millipore, St. Charles, MO, USA), and Mouse Adiponectin ELISA Kit (EMD Millipore, Billerica, MA, USA), respectively. Concentrations of glucose, triglycerides, and cholesterol were measured calorimetrically using Fluitest GLU, Fluitest TG, and Fluitest CHOL (Analyticon^®^ Biotechnologies AG Am Mühlenberg 10, Lichtenfels, Germany), respectively. Concentrations of free fatty acids were measured using NEFA FS DiaSys kits (DiaSys Diagnostic Systems GmbH, Holzheim, Germany). Liver triglycerides were measured as described by Zhao et al. (2014) [40]; liver glycogen was measured by the method of Roehrig and Allred (1974) [41].

### 4.4. Relative Quantitation Real-Time PCR

Total RNA was isolated from tissue samples using ExtractRNA kit (Evrogen, Moscow, Russia) according to the manufacturer’s instructions. First-strand cDNA was synthesized using Moloney murine leukemia virus (MMLV) reverse transcriptase (Evrogen, Moscow, Russia) and oligo(dT) as a primer. TaqMan gene expression assays (Thermo Fisher Scientific, Waltham, MA USA) were used for relative quantitative real-time PCR: Acetyl-coenzyme A carboxylase alpha (*Acca*, Mm01304285_m1); Acetyl-coenzyme A carboxylase beta (*Accb*, Mm01204683_m1); Apolipoprotein B (*Apob*, Mm01545150_m1); Actin, beta as an endogenous control (*Actb*, Mm00607939_s1); Carnitine palmitoyltransferase 1a (*Cpt1a*, Mm01231183_m1); Carnitine palmitoyltransferase 1b (*Cpt1b*, Mm00487191_g1); deiodonase-2 (*Dio2*, Mm00515664_m1); Fatty acid synthase (*Fasn*, Mm00662319_m1); Fibroblast growth factor 21 (*Fgf21*, Mm00840165_g1); Glucokinase (*Gck*, Mm00439129_m1); Glucose-6-phosphatase catalytic subunit (*G6pc*, m00839363_m1); Insulin receptor (*Insr*, Mm01211875_m1); Insulin receptor substrate 1 (*Irs1*, Mm01278327_m1); Insulin receptor substrate 2 (*Irs2*, Mm03038438_m1); Klotho beta (*Klb*, Mm00473122_m1); Lipase hormone sensitive (*Lipe*, Mm00495359_m1), Lipoprotein lipase (*Lpl*, Mm00434764_m1), Patatin-like phospholipase domain containing 2 (*Pnpla2*, Mm00503040_m1); Peroxisome proliferative activated receptor gamma coactivator 1 alpha (*Ppargc1α)*, Mm01208835_m1); Peroxisome proliferator activated receptor alpha (*Pparα*, Mm0040939_m1); Peroxisome proliferator activated receptor gamma (*Pparγ*, Mm00440940_m1); Phosphoenolpyruvate carboxykinase 1, cytosolic (*Pck1*, Mm01247058_m1); Pyruvate kinase liver and red blood cell (*Pklr*, Mm00443090_m1); Solute carrier family 2 member 4 (*Slc2a4*, Mm00436615_m1); Uncoupling protein 1 (*Ucp1*, Mm01244861_m1); and Uncoupling protein 3 (*Ucp3*, Mm01163394_m1). Sequence amplification and fluorescence detection were performed on an Applied Biosystems ViiA 7 Real-Time PCR System (Life Technologies, 5791 Van Allen Way, Carlsbad, CA, USA). Information about the tissues in which gene expression was measured is presented in the Appendix A. Relative quantification was performed by the comparative threshold cycle (CT) method. The mRNA level of the gene in each sample was normalized relative to the average expression value in the control group of males.

### 4.5. Statistical Analysis

Data were analysed using the STAISTICA 10.0 (StatSoft, TIBCO Software Inc., Palo Alto, CA, USA). Descriptive statistics were used. Data are presented as mean ± SE. Comparisons between parameters were made using two-tailed Student’s *t*-test and non-parametric Mann–Whitney U test. Results were considered significant at *p* < 0.05.

## 5. Conclusions

We have shown for the first time that early (pre-pubertal) initiation of an obesogenic diet in males is associated with less pronounced adiposity and greater insulin sensitivity than late (post-pubertal) initiation. In females, in contrast to males, a more effective adaptation was associated with a late start of the diet. In addition, the age of diet initiation influenced not only the adaptation strategy in each sex, but also the manifestation of the sex dimorphism in the adiposity phenotype: early diet initiation was associated with a more favourable phenotype in males, and later in females.

## Figures and Tables

**Figure 1 ijms-25-12436-f001:**
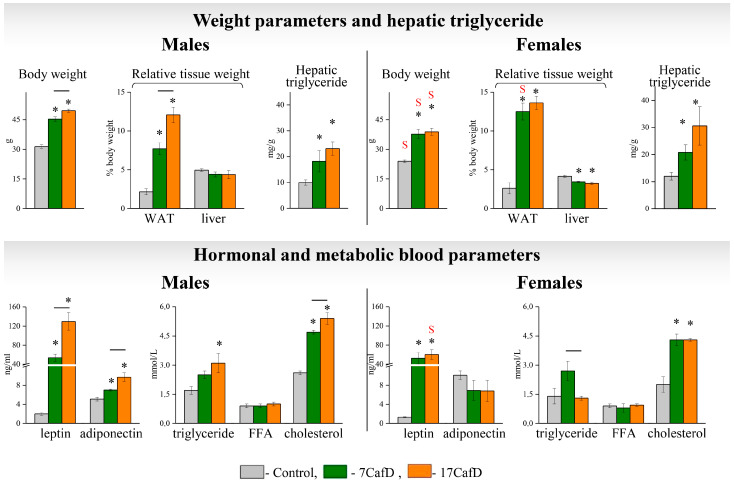
Indicators of body composition and lipid metabolism: body weight (g), relative WAT and liver weight (% body weight), hepatic triglyceride (mg/g), plasma levels of hormonal (leptin and adiponectin, ng/mL), and metabolic (triglycerides, FFAs, and cholesterol, mmol/L) parameters in 27-week-old mice fed standard laboratory chow (control) or cafeteria diet (CafD) from the juvenile age of 7 weeks (7CafD) or from the early adult age of 17 weeks (17CafD). Results are presented as mean ± SE. * *p* < 0.05 vs. control, short line (—) *p* < 0.05 7CafD vs. 17CafD, and ^S^
*p* < 0.05 females vs. males by Student’s *t*-test.

**Figure 2 ijms-25-12436-f002:**
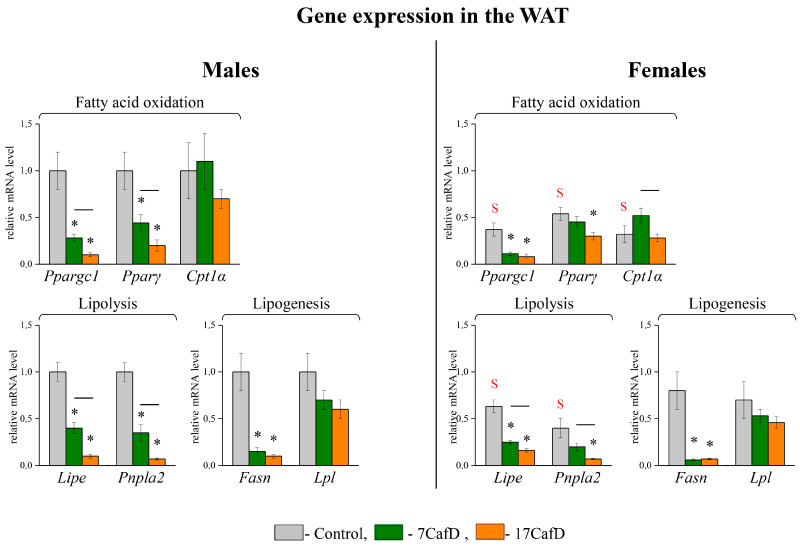
Lipid metabolism. Relative mRNA levels of genes in WAT of 27-week-old mice fed standard laboratory chow (control) or cafeteria diet (CafD) from juvenile age of 7 weeks (7CafD) or early adult age of 17 weeks (17CafD). Results are presented as mean ± SE. * *p* < 0.05 vs. control, short line (—) *p* < 0.05 7CafD vs. 17CafD, and ^S^
*p* < 0.05 females vs. males by Student’s *t*-test.

**Figure 3 ijms-25-12436-f003:**
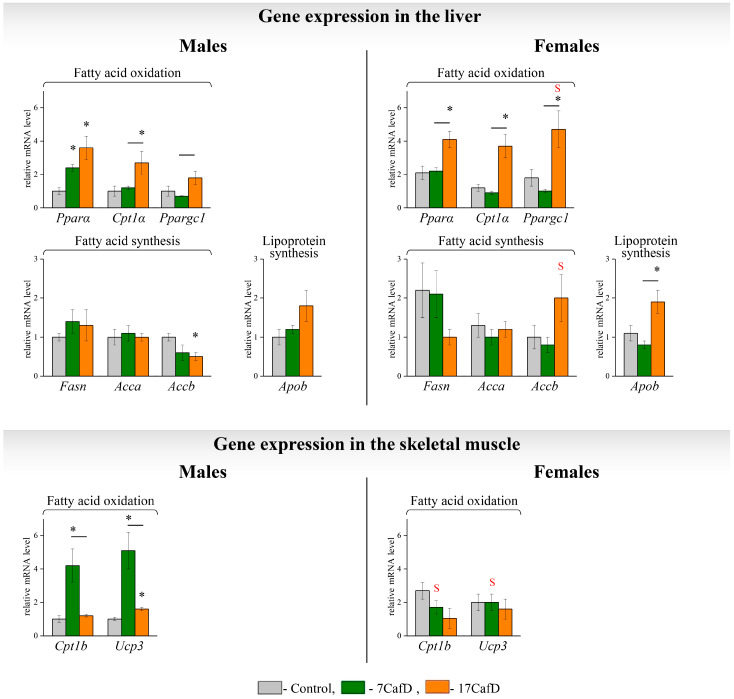
Lipid metabolism. Relative mRNA levels of genes in liver and skeletal muscle of 27-week-old mice fed standard laboratory chow (control) or cafeteria diet (CafD) from the juvenile age of 7 weeks (7CafD) or early adult age of 17 weeks (17CafD). Results are presented as mean ± SE. * *p* < 0.05 vs. control, short line (—) *p* < 0.05 7CafD vs. 17CafD, and ^S^
*p* < 0.05 females vs. males by Student’s *t*-test.

**Figure 4 ijms-25-12436-f004:**
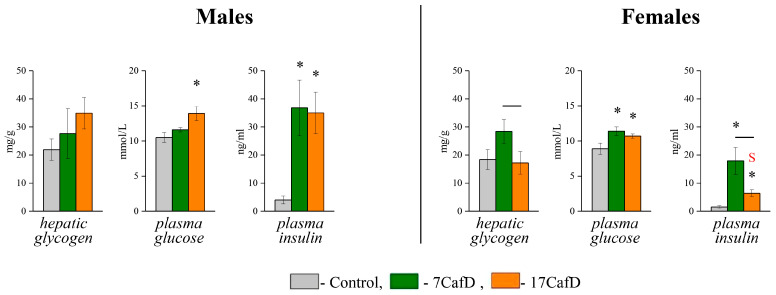
Carbohydrate metabolism. Hepatic glycogen content (mg/g), plasma glucose (mmol/L), and insulin (ng/mL) levels in 27-week-old mice fed standard laboratory chow (control) or cafeteria diet (CafD) from the juvenile age of 7 weeks (7CafD) or early adult age of 17 weeks (17CafD). Results are presented as mean ± SE. * *p* < 0.05 vs. control, short line (—) *p* < 0.05 7CafD vs. 17CafD, and ^S^
*p* < 0.05 females vs. males by Student’s *t*-test.

**Figure 5 ijms-25-12436-f005:**
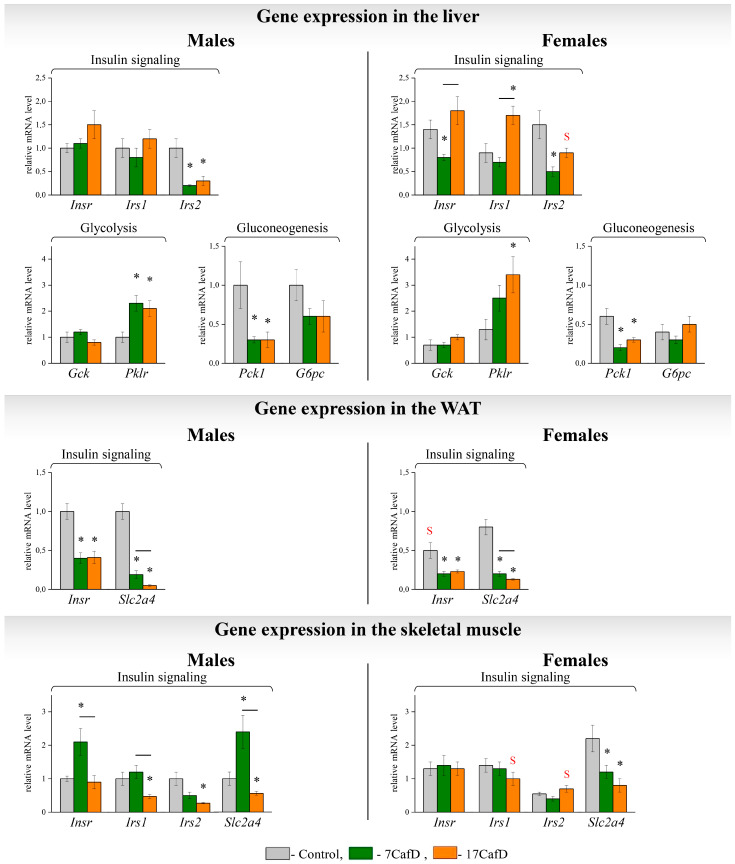
Carbohydrate metabolism. Relative mRNA levels of genes in liver, WAT, and skeletal muscle of 27-week-old mice fed standard laboratory chow (control) or cafeteria diet (CafD) from the juvenile age of 7 weeks (7CafD) or early adult age of 17 weeks (17CafD). Results are presented as mean ± SE. * *p* < 0.05 vs. control, short line (—) *p* < 0.05 7CafD vs. 17CafD, and ^S^
*p* < 0.05 females vs. males by Student’s *t*-test.

**Figure 6 ijms-25-12436-f006:**
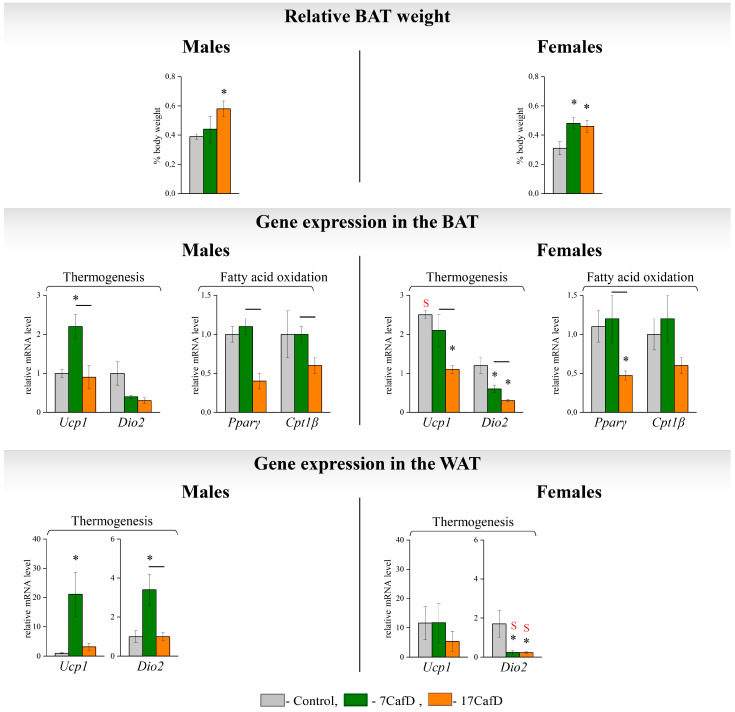
Thermogenesis. Relative BAT weight (% body weight) and mRNA levels of genes in BAT and WAT of 27-week-old mice fed standard laboratory chow (control) or cafeteria diet (CafD) from juvenile age 7 weeks (7CafD) or early adult age 17 weeks (17CafD). Results are presented as mean ± SE. * *p* < 0.05 vs. control, short line (—) *p* < 0.05 7CafD vs. 17CafD, and ^S^
*p* < 0.05 females vs. males by Student’s *t*-test.

**Figure 7 ijms-25-12436-f007:**
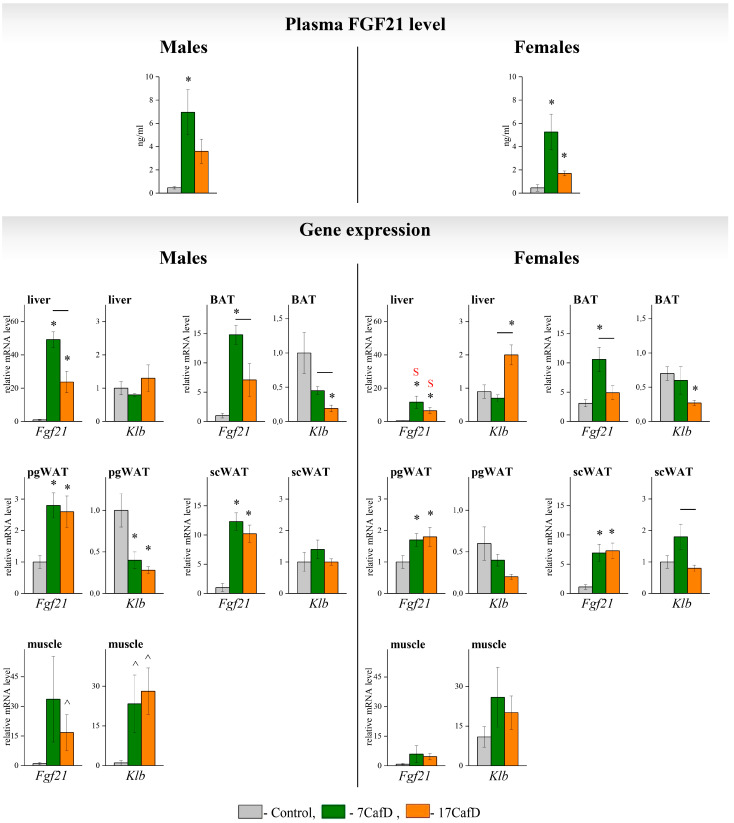
Indicators of FGF21 system activity. Plasma FGF21 levels (ng/mL) (I) and relative mRNA levels of Fgf21 and Klb genes in liver, BAT, WAT, and skeletal muscle (II) of 27-week-old mice fed standard laboratory chow (control) or cafeteria diet (CafD) from the juvenile age of 7 weeks (7CafD) or early adult age of 17 weeks (17CafD). Results are presented as mean ± SE. * *p* < 0.05 vs. control, short line (—) *p* < 0.05 7CafD vs. 17CafD, and ^S^
*p* < 0.05 females vs. males by Student’s *t*-test. ^ *p* < 0.05 vs. control by Mann–Whitney test.

## Data Availability

The original contributions presented in this study are included in the article/Appendix A. Further inquiries can be directed to the corresponding author(s).

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
