# Peer review of "Age of Cafeteria Diet Onset Influences Obesity Phenotype in Mice in a Sex-Specific Manner"

_ijms, 2024, doi:10.3390/ijms252212436_

Round 1

Reviewer 1 Report

Comments and Suggestions for Authors

This paper examines how the timing of exposure of mice to an obesogenic diet alters their metabolism. Mice were fed a standard, control diet, a cafeteria diet from 7 weeks age, or the cafeteria diet from 17 weeks age. All animals reverted to control diet 10 weeks later. Male mice given cafeteria diet from week 7, before puberty, adapted to the cafeteria diet and had normal body weight and white adipose tissue, upregulated thermogenesis adipose genes, greater fatty acid oxidation, and a claimed greater insulin sensitivity than animals commencing the cafeteria diet at 17 weeks. However, females had higher plasma triglyceride levels, hepatic glycogen levels and a greater expression of fatty acid oxida-15 tion genes, and IRS1. The adaptation in male mice at C7 was linked to FGF 21 actions in brown adipose tissue and muscles.

The paper builds on previous publications by the authors showing that the metabolic responses to obesogenic diet differed between young male and female mice (Reference 5). In this previous paper the obesogenic diet was commenced at 10 weeks of age and reported similar metabolic changes between males and females compared to a control diet are shown again here, but with different time points. FGF21 expression was also previously shown to be higher in males than in females in response to the obesogenic diet. The authors should clearly indicate what represents new findings in this paper compared to their previous publications.

The paper looks at multiple variables – diet type, diet timing, sex, multiple tissue types, circulating hormone and metabolite levels, and multiple tissue gene expressions. It is very difficult for the reader to follow this as an emerging pattern of physiological changes. A graphic which tries to show the relationships between all these variables would be very helpful.

In Figure 1 there is no difference between C7 and C17 in body weight at week 27, and no difference between the sexes. C7 males have relatively less white adipose and lower leptin levels than C17 males. Does this mean C7 males gained more lean body mass?

Figure 3 clearly shows that the C7 males much much better fatty acid oxidation than either C17 males of females, but hepatic glycogen is unaltered. This is despite circulation insulin being highly elevated in both C7 and C17 males and females compared to control diet. Why do you think there is no increase in hepatic glycogen storage when insulin is so elevated? The excess carbohydrate in the diet must be going somewhere? In the previous publication (Ref 5)  insulin levels were elevated in the males given obesogenic diet, but not the females, yet the diet in the present study was started only three weeks earlier. This is confusing.

In Figure 5 white adipose tissue INSR and GLUT4 expression are equally suppressed by the obesogenic diet in males and females at both C7 and C17 and in Figure 4 insulin signaling in the liver seems largely unchanged. The only real change is in insulin signaling in skeletal muscle where the receptor expression is greater in males at C7, but IRS expression is suppressed. Overall, would this not equate to similar insulin resistance in most tissues between C7 and C17 in males?

FGF21 levels in the circulation are much greater in males than in females, and higher at C7 than C17. This was shown for the single dietary exposure in the previous paper also.  It is interesting to now compare FGF21 gene expression from liver, adipose tissue and muscle , but clearly this is an order of magnitude higher in liver which will provide the majority of circulating FGF21.

The discussion proposes that FGF21 is a major determinant of the lower adiposity seen in C7 males compared to C17, and compared to females, but this is based only on correlations. A transgenic knock down of hepatic FGF21 expression to test this hypothesis would greatly elevate the value of the paper.

Comments on the Quality of English Language

The grammatical use of English is poor in many places. Perhaps using an AI program might help here.

Reviewer 2 Report

Comments and Suggestions for Authors

The manuscript is focused on the gender differences in responses to age-related different introduction of an obesogenic diet (“cafeteria diet”: energy-dense, hyper-caloric, high carbohydrate, high fat; ad libitum(?)), with regard to metabolic changes and expression of genes involved in metabolism, mainly of carbohydrates and fat, insulin sensitivity, thermogenesis, and fibroblast growth factor 21 (Fgf21) in different tissues (WAT, BAT, liver, muscle, blood). The authors compared C57Bl mice females and males put on the “cafeteria diet” at the age of 7 weeks (before puberty) or age of 17 weeks (after puberty), and controls on standardized diets, until the age of 27 weeks, and found some sexually dimorphic responses.

The value of this manuscript is that numerous important genes were analyzed and presented (however, not all those indicated in the methods), but it was somehow hard to follow the numerous results by graphs. However, the authors managed to summarize the findings in the results and even more in the discussion sections, which helped the readers obtain a more synthesized picture and conclusions, but the author's conclusions were not always supported by the findings (see below).

The main pitfall is that the authors did not implement statistical analyses between genders, but only between different diets among each gender group. Even though from some figures, it was possible to see a clear, substantial difference between genders in the response, for most others it was not clear if this observable difference was statistically significant (having in mind the limited number of animals). 

The discussion and conclusion about the more “favorable” adaptations in males can be questioned by the generally worse biochemistry metabolic profiles in males, compared with females, at the age of 27 weeks, particularly those with the older beginning of the obesogenic diet (after puberty). For the males with the earlier beginning of the obesogenic diet, before puberty, the general biochemistry metabolic profiles were either not different (most of the biochemical parameters), or even worse (e.g., insulin levels), compared with the females. For that reason, authors should base their discussion and conclusions on the more protective (favorable) factors in females after puberty (e.g., the effect of estradiol) that lead to better metabolic adaptations to an obesogenic diet.

Some more specific comments and corrections required (by number of lines):

Abstract:

1.      define a cafeteria diet (composition of macronutrients, or just state “(a diet) containing an increased amount of fats and carbohydrates")

2.      Give the exact number of animals in each dietary intervention group (cafeteria diet vs. control diet). “Cafeteria C57Bl mice starting cafeteria diet from 7-week age (pre-puberty; 9 males, 8 females) or from 17-week age (post-puberty; 7 males, 6 females), and control C57Bl mice on a regular show diet (7 males, 6 females) during 27 weeks“

3.        “up regulating” – “up-regulated”

Introduction

1.      39: “... in adaptation to a cafeteria diet” Please define here what a cafeteria diet is:  cafeteria diet (CafD), a diet containing an increased amount of fats and carbohydrates"

2.      39: “In females, the diet”- which diet? CafD? Please, define

3.      40-46: “induces” – induced, etc. (past tense, only referring to your previous findings, but should not be generalized)

4.      40-46:- which genes were exactly up-regulated/down-regulated in your previous research?

5.      56-60:– define better which age exactly (e.g., how old in weeks they were old in that experiment), and give the reference immediately after “Salinero and co-authors [9] “

Methods

1.      I’m not sure if the journal requires the Methods to be at the end, but I would prefer to have it immediately after the introduction and before the results because it much better introduces the reader to the study design, procedures, and statistical analyses. It is heavy to go back and forth for clarifications in the study design. For this reason, I strongly recommend keeping the standard order of the chapters (Introduction, Methods,...). It is illogical to have methods between discussion and conclusion.

2.      356-357. “Experiments were conducted with С57BL/6J mice housed at the vivarium для  конвенционального разведения of the Institute of Cytology and Genetics, Novosibirsk, Russia.” Please, translate all to English: (...for conventional breeding..)

3.      Give here important information that 7-week-old mice were pre-puberty mice (as you stated in the discussion).

4.      380: “...Energy value 250 kcal/100g kcal.” – give also macronutrient composition of the regular chow diet here,

5.      380: “Cafeteria diet"(CafD): pork lard and biscuits were .. (please, indicate that this is referring to CafD)

6.      Give the energy and macronutrient information of the CafD diet (was this mixed and made as a standard mix?)

7.      Was CafD ad libitum?

8.      Why do not calculate HOMA-IR, as a measure of insulin sensitivity? This information should be very useful, better than glucose and insulin alone.

9.      How was gene expression reported? Per what? Please, give detailed information (additionally, units are missing in the figures).

10.   Why more genes were given in Methods than reported in the results? E.g., Pomc, Acca, Accb, Apob,...many of them were mentioned in methods, but not given in the results. Please, delete them.

11.   Statistical analyses: The statistics are missing for the gender differences in relative (percentage) or absolute increases/decreases in the studied parameters between 7CafD males and females, and between 17CafD males and females (e.g., T-test/Man Whitney test between males and females with increases/decreases expressed as % change compared to the standardized diet, or ANCOVA should be applied, e.g., for SPSS: https://statistics.laerd.com/spss-tutorials/ancova-using-spss-statistics.php#procedure). The results can be given as Supplementary tables.

Results:

1.      Referring to the previous lines, what about gender differences in relative (percentage) increase in body weight, WAT, hepatic triglycerides, leptin, adiponectin, FFA, TG, and cholesterol (Figure 1) between 7CafD males and females, and between 17CafD males and females? The same for the gender differences in the changes in all other examined parameters in Figures 2 -7 (e.g., ANCOVA should be applied).

Give the information in the text with the exact P values for each of these significant differences between males and females (for both 7-week and 17-week), since they are not shown in Figures 1-7, only differences between 3 groups among each sex.

2.      Figures (all): please, give the units on the Y-axis, and information on how the way gene expression was calculated.

3.      I strongly suggest that Figure 4 goes immediately after Figure 1, since they are both associated with metabolic parameters (weight measurements, biochemistry,..), , and that all figures with gene expression go after this.

4.      Interestingly, Figures 1 and 4 suggest much better metabolic profiles in obese females compared to males, which is important for discussion and conclusion. (e.g., glucose, TG, CHOL, insulin, leptin, ..)

5.      Figure 2. Give the title “Gene expression in the WAT” (as you did for other figures)

6.      74-75: “In females kept on a diet,” – which diet? CafD? please specify “... kept on both CafD diets (7CafD and 17CafD)”

7.      98: “Fasn, Lipe, and Pnpla2 was lower than in the control. “ – this should not be a new paragraph, but a continuation of line 89.

8.      202-203: “The hormone level in the blood of 7CafD females was 3 times higher than that of 7CafD females (tendency P < 0.06).”- ??, please check; 17CafD?

Discussion

1.      As I said, the discussion and conclusion about the more “favorable” adaptations in males can be questioned by the generally “worse metabolic profiles” in those males, compared with females, or “no different metabolic profiles”, regarding the initiation time of the obesogenic diet.

2.      256-258: “The effect of FGF21 on metabolic phenotype is partially mediated by its effect on gene expression in metabolic tissues [16] [17]. “ – Which genes and which metabolic tissues? Please, give some more details on it.

3.      271-272: “Pharmacological administration of FGF21 is known to enhance the expression of FAO genes in muscle [20] and increase insulin sensitivity [21–23].” – in your study, you found an increase in insulin receptor transcription and glucose transporter in muscles, but the other genes involved in this receptor functions (IRS1 and 2) were not increased, which can be explained by the compensatory try to increase the function of the receptor, but probably without effect, since insulin levels were still increased in 7CafD males (indicating insulin resistance). HOMA-IR will this show even more.

4.      277-279: “We suggest that the effect of age of diet initiation on the efficiency of adaptation is partly due to differences in the activation of the FGF21 system, but many other mechanisms could be also involved.” (Please, do not try to exclude the other possible mechanisms, not studied here!). Additionally, your study did not show the direct effect of FGF21 on insulin sensitivity and FAO, and all findings can also be independent. So the conclusion of the possible mechanisms involving only the FGF21 system is very limiting.

5.      280-316: how is this related to hepatic TG content in females? (It seems that in 7CafD females is in the middle between standard diet and 17CafD females, even though it was not significant, and LP secretion was not increased) – this should be more discussed.

6.      Please, give more about the effects of sex hormones on these metabolic adaptations (direct or indirect), (testosterone, not only estradiol) and genes associated with X or Y chromosomes. There is plenty of literature on it.

7.      Did, in fact, estrogens after puberty counteract the negative effects of the obesogenic diets in females, providing the final result in generally much better metabolic profiles in obese females compared to males, particularly when the obesogenic diet was started after puberty? Your study shows that in mice that start the obesogenic diet before puberty, there is no difference in most of the biochemical parameters between males and females

8.      339: “The cafeteria diet is the most adequate model for the development of diet-induced obesity in humans. “ – please, delete “the most”, the cafeteria diet is only one of the models for the development of diet-induced obesity in humans.

9.      Give study limitations. The smaller number of animals in certain subgroups for statistical analyses can be also the question (limitation).

10.   342: “While in humans, diets that lead to obesity are more typical for adults. “ – an unclear and unfinished sentence, please delete it.

Conclusion

1.      The whole conclusion should be changed in line with necessary additional statistical analyses for the differences between genders in each diet group, and what was said on the generally worse metabolic profiles in males that started an obesogenic diet after puberty, compared with females, while no significant metabolic differences observed between males and females if the obesogenic diet had stared before puberty (see Figures 1 and 4).

Round 2

Reviewer 1 Report

Comments and Suggestions for Authors

The authors have revised the paper taking into account the reviewer comments. It has been improved.

Author Response

Thank you very much for your attention to our manuscript.

Reviewer 2 Report

Comments and Suggestions for Authors

Dear authors and editor,

There are significant improvements, but small changes should be further made.

1.       The T-Test (or Mann-Whitney) should be given for % differences from control animals. For example: in 7 CadfD males there was a 10% increase in leptin compared to control males, while in 7 CadfD females, there was a 30% increase compared to control females (p=0.001), while in 17 CadfD males, there was an 8% increase in leptin compared to control animals, while in female 17 CadfD animals there was a 15% increase compared to control animals (p=0.056) (This p is related to gender differences for increase percentages).  Therefore, you should compare the % change (increase or decrease) above control animals of the same gender, because there are differences between genders even in control animals, which you show with your analyses when gender-comparing control animals. You can keep S for differences in absolute terms if you want, but make additional calculations for % increase/decrease above the control animals of the same gender, this is more important for your study.

For example, in the article https://pmc.ncbi.nlm.nih.gov/articles/PMC8511067/#S8

To account for sex and individual variability, wheel running and body weight were all analyzed as percent baseline.” 

The more correct is to use ANCOVA for such calculations, but it is OK to calculate differences (%), and then compare them by T-test or Man-Whitney (depending on distribution)

2.       In the figure legend, it should be stated what S means, and the symbol you will use for the gender differences in percentage increase /decrease should be explained.

3. The abstract, discussion, and conclusion will be changed after making those additional calculations.

4.       Abstract: Please provide the number of animals, correct English, summarize (eliminate the repetition), and reframe.

Here is an example of the shorter version with all the information given (200 words):

Abstract:

We investigated the influence of sex and age of an obesogenic diet initiation on the later-age obesity phenotypes. C57Bl mice started the Cafeteria Diet (CafD, with increased fat and carbohydrates, ad libitum) from 7-week age (7CafD, pre-puberty) or 17-week age (7CafD, post-puberty) while control C57Bl mice were fed regular chow. At 27 weeks of age, 7CafD males (n=9) compared to 17CafD males (n=7) had lower body weight, white adipose tissue (WAT) relative weight, plasma cholesterol levels, and higher expression of thermogenic genes in WAT and brown adipose tissue (BAT), and fatty acid oxidation (FAO) and insulin signaling genes in muscles. 7CafD females (n=8) compared to 17CafD females (n=6) had higher plasma triglyceride levels and hepatic glycogen content, but lower insulin sensitivity and hepatic expression of FAO and insulin signaling genes. 7CafD females compared to 7CafD males had more WAT, and reduced expression of FAO genes in muscles and thermogenic genes in WAT. 17CafD females compared to 17CafD males had lower plasma leptin and insulin levels, higher insulin sensitivity, and expression of insulin signaling genes in the liver and muscles. Thus, initiation of the obesogenic diet before puberty led to more adaptive metabolic phenotypes in males, and after puberty – in females.

English language corrections:

64: We reported that in C57Bl/6J male and female mice differ in adaptation to a cafeteria diet (CafD), a diet containing an increased amount of fats and carbohydrates

82-96: Please be free to use the standard scientific abbreviations: HFD (high-fat diet) and DIO (diet-induced obesity). It is unnecessary to eliminate them, and be sure that the abbreviations are defined.

83: Salinero and co-authors [9] ..... (give reference in that sentence after giving the name of authors)

83-90: please, use the past tense (“were” not “are”, “had” not “have”). You only refer to the findings of Salinero and co-authors, what they have found (which should not be generalized, and your findings are the opposite).

100. Use the abbreviation CafD instead of “the cafeteria diet”

101, 104. “The weights of body, white adipose tissue (WAT)“ unusual for English and incorrect, instead say more correctly: “Body weight, relative white adipose tissue (WAT) weight,

Probably there are some other English corrections needed, but it is difficult to follow the text with both insertions and deletions shown. Probably, when uploading the corrected version, it is better to hide deletions (only insertions make visible, and label with yellow the corrected sentences)

Comments on the Quality of English Language

Probably there are more corrections needed, it is difficult to follow insertions and deletions

Author Response

Point-by-point response to Comments and Suggestions for Authors

Reviewer 2

Comments 1, 2, 3: The T-Test (or Mann-Whitney) should be given for % differences from control animals. For example: in 7 CadfD males there was a 10% increase in leptin compared to control males, while in 7 CadfD females, there was a 30% increase compared to control females (p=0.001), while in 17 CadfD males, there was an 8% increase in leptin compared to control animals, while in female 17 CadfD animals there was a 15% increase compared to control animals (p=0.056) (This p is related to gender differences for increase percentages).  Therefore, you should compare the % change (increase or decrease) above control animals of the same gender, because there are differences between genders even in control animals, which you show with your analyses when gender-comparing control animals. You can keep S for differences in absolute terms if you want, but make additional calculations for % increase/decrease above the control animals of the same gender, this is more important for your study.

For example, in the article https://pmc.ncbi.nlm.nih.gov/articles/PMC8511067/#S8

To account for sex and individual variability, wheel running and body weight were all analyzed as percent baseline.” 

The more correct is to use ANCOVA for such calculations, but it is OK to calculate differences (%), and then compare them by T-test or Man-Whitney (depending on distribution)

  1. In the figure legend, it should be stated what S means, and the symbol you will use for the gender differences in percentage increase /decrease should be explained.
  2. The abstract, discussion, and conclusion will be changed after making those additional calculations.

Responses 1, 2, 3: We agree with you that measuring the response as a percentage provides additional ways to compare the strength of the response. To statistically compare the response rates in the 7CafD and 17CafD groups, it is necessary to measure them at baseline before the intervention (in our case, before the diet) and at the endpoint for each animal, calculate the mean and its error, and then compare the samples. In our case (except for body weight) all parameters (hormone and metabolite levels, tissue weight, gene expression) were measured once in animals at the end of the experiment after decapitation. Control animals are different from DIO animals. For animals with DIO, the parameters of control mice cannot be used as a baseline. We were therefore unable to make a statistical comparison of the percentages of response and include the data in the Results and Discussion.  Thank you for the link to an interesting article. //pmc.ncbi.nlm.nih.gov/articles/PMC8511067/#S8. The authors of this article measured the percentage change in only two parameters: body weight and wheel running. That is, parameters that could be measured at two points – the beginning and the end of the experiment.

Comment 4: Abstract: Please provide the number of animals, correct English, summarize (eliminate the repetition), and reframe.

We investigated the influence of sex and age of an obesogenic diet initiation on the later-age obesity phenotypes. C57Bl mice started the Cafeteria Diet (CafD, with increased fat and carbohydrates, ad libitum) from 7-week age (7CafD, pre-puberty) or 17-week age (7CafD, post-puberty) while control C57Bl mice were fed regular chow. At 27 weeks of age, 7CafD males (n=9) compared to 17CafD males (n=7) had lower body weight, white adipose tissue (WAT) relative weight, plasma cholesterol levels, and higher expression of thermogenic genes in WAT and brown adipose tissue (BAT), and fatty acid oxidation (FAO) and insulin signaling genes in muscles. 7CafD females (n=8) compared to 17CafD females (n=6) had higher plasma triglyceride levels and hepatic glycogen content, but lower insulin sensitivity and hepatic expression of FAO and insulin signaling genes. 7CafD females compared to 7CafD males had more WAT, and reduced expression of FAO genes in muscles and thermogenic genes in WAT. 17CafD females compared to 17CafD males had lower plasma leptin and insulin levels, higher insulin sensitivity, and expression of insulin signaling genes in the liver and muscles. Thus, initiation of the obesogenic diet before puberty led to more adaptive metabolic phenotypes in males, and after puberty – in females.

Response 4: We are very grateful to you for not only providing useful comments, but also for really helping us to shape the article. We have used the version of the Abstract that you suggested.

Comments 5: English language corrections:

64: We reported that in C57Bl/6J male and female mice differ in adaptation to a cafeteria diet (CafD), a diet containing an increased amount of fats and carbohydrates

Response 5: This correction has been made

Comments 6: 82-96: Please be free to use the standard scientific abbreviations: HFD (high-fat diet) and DIO (diet-induced obesity). It is unnecessary to eliminate them, and be sure that the abbreviations are defined.

Response 6: Thank you for your comments. We have made the appropriate changes to the text.

Comment 7: 83: Salinero and co-authors [9] ..... (give reference in that sentence after giving the name of authors)

Response 7: It was done.

Comments 8: 83-90: please, use the past tense (“were” not “are”, “had” not “have”). You only refer to the findings of Salinero and co-authors, what they have found (which should not be generalized, and your findings are the opposite).

Response 8: We used the past tense in this paragraph.

Comment 9: 100. Use the abbreviation CafD instead of “the cafeteria diet”

Response 9: It was done.

Comment 10: 101, 104. “The weights of body, white adipose tissue (WAT)“ unusual for English and incorrect, instead say more correctly: “Body weight, relative white adipose tissue (WAT) weight,

Response 10: Thank you again for your kind advice and help. We have made these changes to the text.